# Dielectric Responses of (Zn_0.33_Nb_0.67_)*_x_*Ti_1−*x*_O_2_ Ceramics Prepared by Chemical Combustion Process: DFT and Experimental Approaches

**DOI:** 10.3390/molecules27186121

**Published:** 2022-09-19

**Authors:** Theeranuch Nachaithong, Pairot Moontragoon, Prasit Thongbai

**Affiliations:** 1Materials Science and Nanotechnology Program, Faculty of Science, Khon Kaen University, Khon Kaen 40002, Thailand; 2Institute of Nanomaterials Research and Innovation for Energy (IN-RIE), Khon Kaen University, Khon Kaen 40002, Thailand; 3Thailand Center of Excellence in Physics, Commission on Higher Education, Bangkok 10400, Thailand; 4Department of Physics, Faculty of Science, Khon Kaen University, Khon Kaen 40002, Thailand

**Keywords:** TiO_2_, electron hopping, dielectric constant, loss tangent, DFT

## Abstract

The (Zn, Nb)-codoped TiO_2_ (called ZNTO) nanopowder was successfully synthesized by a simple combustion process and then the ceramic from it was sintered with a highly dense microstructure. The doped atoms were consistently distributed, and the existence of oxygen vacancies was verified by a Raman spectrum. It was found that the ZNTO ceramic was a result of thermally activated giant dielectric relaxation, and the outer surface layer had a slight effect on the dielectric properties. The theoretical calculation by using the density functional theory (DFT) revealed that the Zn atoms are energy preferable to place close to the oxygen vacancy (Vo) position to create a triangle shape (called the ZnVoTi defect). This defect cluster was also opposite to the diamond shape (called the 2Nb2Ti defect). However, these two types of defects were not correlated together. Therefore, it theoretically confirms that the electron-pinned defect-dipoles (EPDD) cannot be created in the ZNTO structure. Instead, the giant dielectric property of the (Zn_0.33_Nb_0.67_)*_x_*Ti_1__−*x*_O_2_ ceramics could be caused by the interfacial polarization combined with electron hopping between the Zn^2+^/Zn^3+^ and Ti^3+^/Ti^4+^ ions, rather than due to the EPDD effect. Additionally, it was also proved that the surface barrier-layer capacitor (SBLC) had a slight influence on the giant dielectric properties of the ZNTO ceramics. The annealing process can cause improved dielectric properties, which are properties with a huge advantage to practical applications and devices.

## 1. Introduction

Recently, giant or colossal dielectric permittivity oxide materials, which are the metal oxides with a high dielectric permittivity (ε′) (more than 10^3^), e.g., BaTiO_3_, CCTO, LaFeO_3_ and codoped NiO-based oxides [1,2,3,4,5,6,7], have been extensively researched [8,9,10,11,12,13,14,15]. They have a potential to apply in many devices, i.e., multilayer ceramic capacitors and high-energy-dense storage devices. However, there are still some practical problems about their high dielectric loss tangent (tanδ) values, particularly in a range of low frequency.

In recent years, the giant dielectric properties of (In, Nb)-codoped TiO_2_ ceramics (called INTO) [16] have been studied. It has been reported that codoped TiO_2_ ceramics show a very high ε′ (~10^4^), low tanδ (~0.02) and high temperature stability from 80 to 450 K. However, there is still controversy in the origination of this colossal dielectric behavior. In some reports, the electron-pinned defect-dipole (EPDD) was caused by the colossal permittivity of this metal oxide. On the other hand, it was claimed that the colossal permittivity was contributed from the other effects, such as the internal barrier-layer capacitor (IBLC) effect, the surface barrier-layer capacitor (SBLC), the sample–electrode contact effect and polaronic hopping models [16,17,18,19,20]. Therefore, in this work, the mechanics inside of the giant dielectric permittivity of TiO_2_-based ceramics will be reported and clarified. Although the colossal dielectric response of (Zn, Nb)-codoped TiO_2_ was widely investigated [21,22], the colossal permittivity properties of these system ceramics synthesized by wet chemical routes has not been widely reported. Thus, in this study, both experimental and theoretical studies were simultaneously performed on TiO_2_ ceramics codoped with Zn^2+^/Nb^5+^ ions. In the experimental part, both the (Zn_0.33_Nb_0.67_)*_x_*Ti_1−*x*_O_2_ powders and sintered ceramics were thoroughly fabricated and characterized. Their colossal permittivity properties at different frequencies and temperatures were investigated to expose the mechanics and behavior inside the microstructures. In the theoretical aspect, the density functional theory (DFT) was employed to evaluate the ground-state properties of the (Zn_0.33_Nb_0.67_)*_x_*Ti_1−*x*_O_2_ to understand the cause of the colossal permittivity in the (Zn, Nb)-codoped TiO_2_ ceramics. The feasible explanation for the noticed giant permittivity behavior will be reported.

## 2. Experimental Details

(Zn_0.33_Nb_0.67_)*_x_*Ti_1−*x*_O_2_ (ZNTO) nanoparticles (*x* = 0.5, 1, 2.5, 5.0 and 10%) were synthesized by a simple combustion method. C_16_H_28_O_6_Ti (Sigma-Aldrich, Bangkok, Thailand), NbCl_5_ (Sigma-Aldrich, >99.9%), N_2_O_6_Zn·*x*H_2_O (Sigma-Aldrich, 99.999%) were weighed corresponding to each doped content. Firstly, NbCl_5_ and N_2_O_6_Zn·*x*H_2_O were mixed in citric acid and then a C_16_H_28_O_6_Ti solution was mixed at 130 °C and stirred until a gel was obtained. Secondly, all doping concentrations gels were calcined, then they were pressed into pellets with diameter of 9.5 mm and thickness of ~1.3 mm. Finally, they were sintered at 1400 °C for 5 h. (Zn_0.33_Nb_0.67_)*_x_*Ti_1−*x*_O_2_ ceramics with *x* = 0.5 and 0.25% are referred to as the 0.5% ZNTO and 2.5% ZNTO ceramics, respectively.

The phase structure of all samples was characterized by X-ray diffraction technique (XRD, PANalytical, EMPYREAN). The field emission scanning electron microscopy technique (FE-SEM, FEI, Hileos Nanolab G3CX) was performed to reveal the surface morphologies of the homogeneity of all elements in the sintered samples. To confirm existing oxygen vacancies in the microstructures, the Raman spectra of sintered ceramics were measured with a Raman System (NT-MDT Ntegra Spectra), using laser wavelength of 532 nm. Finally, the dielectric properties or permittivity responses were measured by an impedance analyzer technique (KEYSIGHT E4990A). In this study, the temperature dependence of the dielectric constant (ε′) and loss tangent (tanδ) was measured in the frequency and temperature ranges of 10^2^–10^6^ Hz and −60 to 200 °C. In order to thoroughly understand inside the nanoscale level, the stable configuration of the periodic boundary conditions of 2 × 2 × 6 super-cell Zn and Nb-codoped rutile–TiO_2_ structure was studied using the DFT calculation, performed under Vienna Ab initio Simulation Package (VASP) with Projector-augmented plane-wave pseudopotential method (PAW) and the Perdew–Burke–Ernzerhof (PBE) form of exchange–correlation functional. In this model, firstly, one oxygen vacancy was created and then replaced two Ti atoms with Zn atoms to form 2ZnVo triangular defect and another two Ti atoms with Nb atoms to form 2NbTi diamond defect. Additionally, cutoff energy of 600 eV and 3 × 3 × 3 k-point meshes in Monkhorst–Pack k-point are also employed to optimize the structures to obtain the lowest energy-preferable configuration, the conjugate-gradient algorithm and Hellmann–Feynman theorem were carried out to calculate the force acting on each ion and 5 × 5 × 5 k-point meshes in Monkhorst–Pack k-point are used to calculate the electronic structures. The orbitals of Ti(3p^6^ 4s^2^ 3d^2^), Zn(3p^6^ 4s^2^ 3d^10^), O(2s^2^ 2p^4^) and Nb(4p^6^ 5s^1^ 4d^4^) were treated as valence electrons.

## 3. Results and Discussion

According to Figure 1, the XRD patterns of the ZNTO powders prepared by a chemical combustion method confirmed the rutile-TiO_2_ (JCPDS 21-1276) [23,24] phase and no impurity phase, which is in good agreement with other works [23,24,25]. The *a* and *c* values, extracted from the XRD patterns, are shown in Table 1, and they increased with the rising doping content. These results confirmed that the Zn^2+^ and Nb^5+^ can substitute in the Ti sites of the rutile structure of TiO_2_. The increase in the lattice parameters was due to the larger ionic radii of the dopants compared to that of the Ti^4+^ host ion.

According to Figure 2, the surface morphologies of the ZNTO ceramic sintered at 1400 °C for 5 h with 0.5 and 2.5% were revealed. It shows that the grains with grain sizes of 5.7 ± 2.4 and 3.7 ± 1.3 µm, respectively, and the grain boundaries were clearly observed. The microstructure of the sintered ceramics was highly dense without pores in the 0.5% ZNTO and 2.5% ZNTO ceramics. The mean grain size of the ZNTO ceramics decreased with an increasing codoping concentration. This result is similar to those reported in the literature by Nachaithong et al. [26] and Yang et al. [27]. It was explained that the decreased mean grain size of the codoped TiO_2_ ceramics was caused by the solute drag mechanism.

As shown in Figure 3, the Raman spectra of the pure TiO_2_ and ZNTO ceramics with the E_g_ and A_1g_ modes were presented. The E_g_ peaks of the TiO_2_, 0.5% Nb-doped TiO_2_, 0.5% Zn-doped TiO_2_, 0.5% ZNTO and 2.5% ZNTO samples appeared at 446.5, 446.5, 446.5, 444.5 and 444.5 cm^−1^, respectively, whereas the A_1g_ modes appeared at 610.5, 610.0, 611.0, 611.5 and 610.5 cm^−1^, respectively. It was shown that the oxygen vacancies and O–Ti–O bonds were associated with the E_g_ and A_1g_ modes [28].

According to Figure 4, the SEM-mapping method was used to reveal a distribution of the elements in the ZNTO ceramics (such as Zn, Nb, Ti and O), and it was found that there is a homogeneous dispersion of the doped elements in the microstructure.

Figure 5 shows the ε′ permittivity at room temperature of the Pure-TiO_2_, 0.5% ZNTO and 2.5% ZNTO ceramics. All the samples exhibited a giant dielectric permittivity in the frequency range of 40–10^6^ Hz. The ε′ value of a pure-TiO_2_ was the lowest. Although a rutile-TiO_2_ had the largest ε′ value among the simple oxides, its ε′ value is very low compared to those of many complex oxides, such as BaTiO_3_ and CaCu_3_Ti_4_O_12_-based materials. Nevertheless, the ε′ value of TiO_2_ can be significantly enhanced by codoping metal ions. In the tanδ spectrum, the samples showed that the tanδ peaks in the frequency range of 10^3^–10^4^ and ε′ slightly changed. This dielectric relaxation behavior could imply that the Nb^5+^ and Zn^2+^ codoping ions have slightly affected the ionic polarization. However, when it was considered in a low-frequency range, the tanδ increased with an increasing codopant concentration. It shows that the codopants have an effect on the interfacial polarization which is usually induced at the internal insulating layer, i.e., a semiconducting region or surface of the sample–electrode layer of ceramics. The ε′ values of the 0.5% ZNTO and 2.5% ZNTO ceramics were about ≈ 9 × 10^4^ and 3 × 10^4^ with tanδ ≈ 0.26 and 1.25 at 1 kHz and RT, respectively.

To study the effects of the surface on the electrical properties of ZNTO ceramics in Figure 6, after the dielectric properties of the as-fired sample were measured, both sides of the electrodes and the outer surface layers of the pellet were removed by polishing them with SiC paper (referred to as the polished-sample), and after that, the polished-sample was measured, the electrodes were removed by the SiC paper and annealed at 1200 °C in air for 30 min (referred to as the annealed-sample). In comparison with the as-sample, polished-sample and annealed-sample, at room temperature, all the samples exhibited very high ε′ values of ≈10^4^–10^5^. The change in the ε′ and tanδ was not considerable compared with that of the polished-sample, but the dielectric permittivity slightly increased in frequency, more than 10^3^ Hz. The tanδ decreased significantly when compared with the as-sample. After the annealing process, the greatly reduced tanδ value and greatly increased ε′ value in the anneal-sample were primarily due to the filling of the oxygen vacancies on the surface. Therefore, the SBLC effect was a key factor for the anneal-sample. In this experiment, it was clearly shown that the outer surface layer influences the dielectric responses of the ZNTO ceramics. The annealing method was suggested to be one of the most important for the improvement of the dielectric properties of ZNTO ceramics by creating a resistive outer surface layer. Furthermore, this method could be a new outer surface design approach for other codoped TiO_2_ ceramic systems.

To explain the behavior of the colossal dielectric properties in ZNTO ceramics, the effect of temperature on the dielectric behavior at different frequencies and temperatures was investigated, as shown in Figure 7. It is seen that the ε′ step likely decreases with an increasing frequency, but increases with an increasing temperature, corresponding to a decrease in the resistance (R). Moreover, the corresponding tanδ peak also increases. This confirmed the thermally activated giant dielectric relaxation behavior. According to Figure 8, the Arrhenius plots of the 0.5% ZNTO and 2.5% ZNTO ceramics show the activation energy (E_a_) which is the temperature dependence of the relaxation peak of the dielectric loss (f_max_), following the Arrhenius law:(1)$$f_{\max}{=f}_{0}\exp\left(\frac{{-E}_{a}}{K_{B}T}\right)$$

It is required energy for the dielectric relaxation in the ceramics and calculated from the slope of the ln f_max_ vs. 1000/T plots. The E_a_ values of the 0.5% ZNTO and 2.5% ZNTO ceramics were 0.213 and 0.175 eV., respectively. It is implied that the giant dielectric response could be related to the Zn^2+^/Zn^3+^ or Ti^3+^/Ti^4+^ electron hopping. Increasing the Nb^5+^ doping concentration could result in a Ti^3+^/Ti^4+^ ions ratio increase. The concentration of the free electrons in the Nb^5+^-doped TiO_2_ is generally proportional to the Nb^5+^ dopant concentration, following the equations [16]:(2)$${2\mathrm{TiO}}_{2}{+\mathrm{Nb}}_{2}O_{5}\overset{{4\mathrm{TiO}}_{2}}{\to }{2\mathrm{Ti}’}_{\mathrm{Ti}}{+2\mathrm{Nb}}_{\mathrm{Ti}}^{•}{+8O}_{O}+\frac{1}{2}O_{2},$$
(3)$${\mathrm{Ti}}^{4+}+e\to {\mathrm{Ti}}^{3+}$$

Therefore, the electron hopping can be readily stimulated when the Ti^3+^ or Zn^3+^ content rises, giving rise to a decrease in the E_a_ values for the ZNTO ceramics with *x* = 0.5 and 2.5%.

According to the theoretical investigations, we placed 2Nb atoms preferentially forming a diamond-shaped structure (called 2Nb2Ti defect) and the 2ZnVo triangular defect into the TiO_2_ structure simultaneously in three configurations: near, opposite and far. The lowest total energy can be obtained when the 2Nb diamond defects and 2ZnVo triangular defects are opposite (as shown in Figure 9), corresponding to the EPDD model and not in the ZNTO structure. It can be clearly suggested that the giant dielectric relaxation behavior of the ZNTO ceramics did not originate from the EPDD effect. Moreover, the electron hopping mechanism between the Ti^3+^/Ti^4+^ ions and Zn^2+^/Zn^3+^ is also the most likely mechanism to be related to the giant dielectric relaxation in the ZNTO ceramics.

## 4. Conclusions

In this work, the (Zn + Nb)-codoped TiO_2_ ceramics prepared by a combustion process show a colossal dielectric permittivity and a low loss behavior. Moreover, their dielectric characteristics show a good frequency stability. The annealing process can cause improved dielectric properties, which are properties with a huge advantage in practical applications and devices. In addition, the giant dielectric relaxation behavior of the ZNTO originated from both the IBLC and the Ti^3+^/Ti^4+^ ions and Zn^2+^/Zn^3+^ electron hopping mechanism. It indicates that there is giant dielectric relaxation behavior in the (Zn, Nb)-codoped TiO_2_ systems.

## Figures and Tables

**Figure 1 molecules-27-06121-f001:**
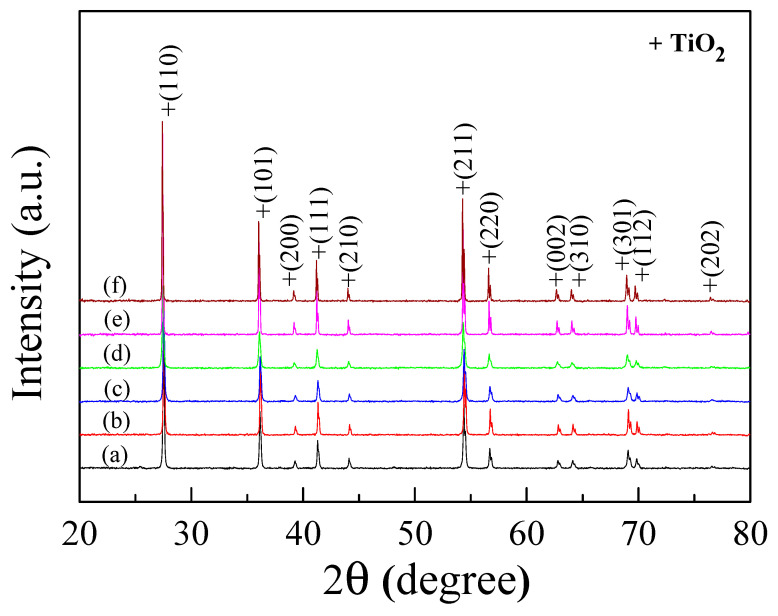
XRD patterns of single doped (a) Zn, (b) Nb in TiO_2_ powder and (Zn_0.33_Nb_0__.67_)*_x_*Ti_1−*x*_O_2_ powder with *x* = (c) 0.5%, (d) 2.5% and (e) 0.5% ZNTO, (f) 2.5% ZNTO ceramics.

**Figure 2 molecules-27-06121-f002:**
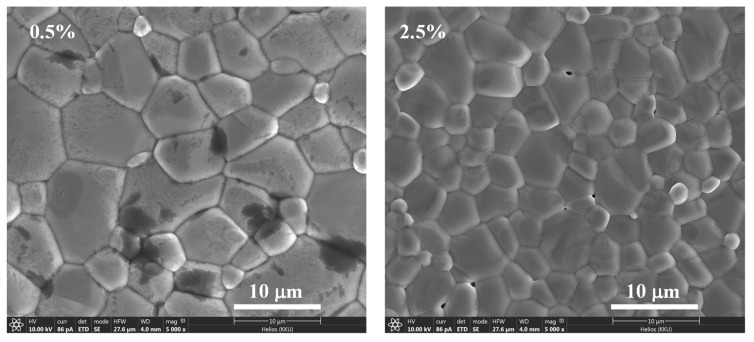
Surface morphologies of the (Zn_0.33_Nb_0.67_)*_x_*Ti_1−*x*_O_2_ ceramics with *x* = 0.5 and 2.5%.

**Figure 3 molecules-27-06121-f003:**
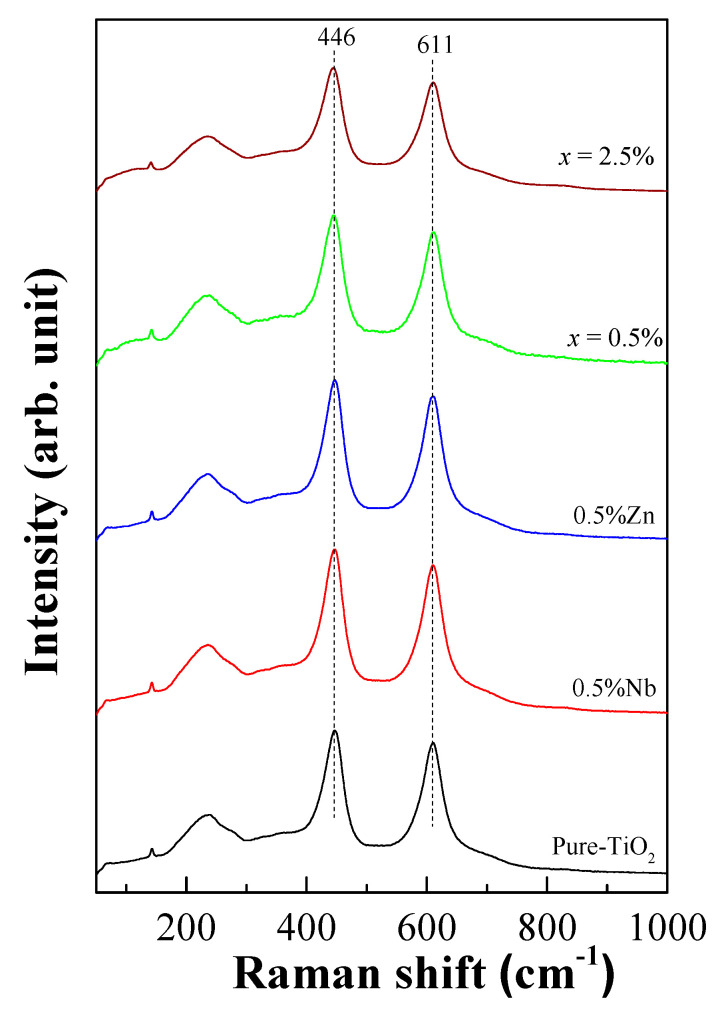
Raman spectra of rutile-TiO_2_, single doped of Nb^5+^ and Zn^2+^ and ZNTO ceramics with difference codoping levels.

**Figure 4 molecules-27-06121-f004:**
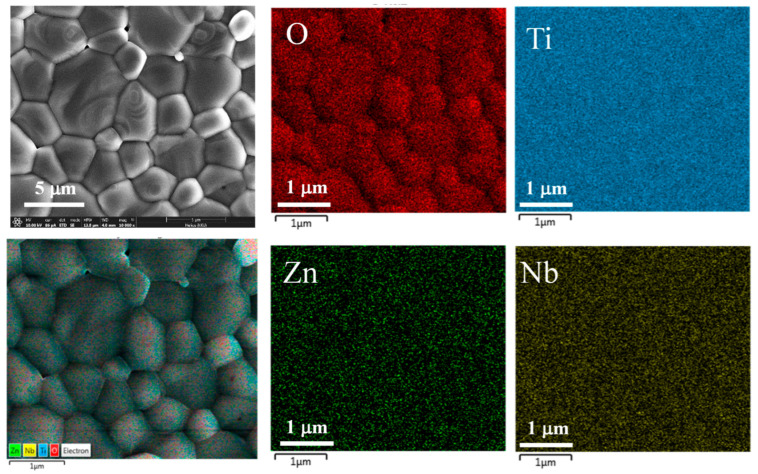
The elements mapping of 2.5% ZNTO ceramic; the Zn, Nb, Ti and O dopants are homogeneously dispersed in the grains and grain boundaries.

**Figure 5 molecules-27-06121-f005:**
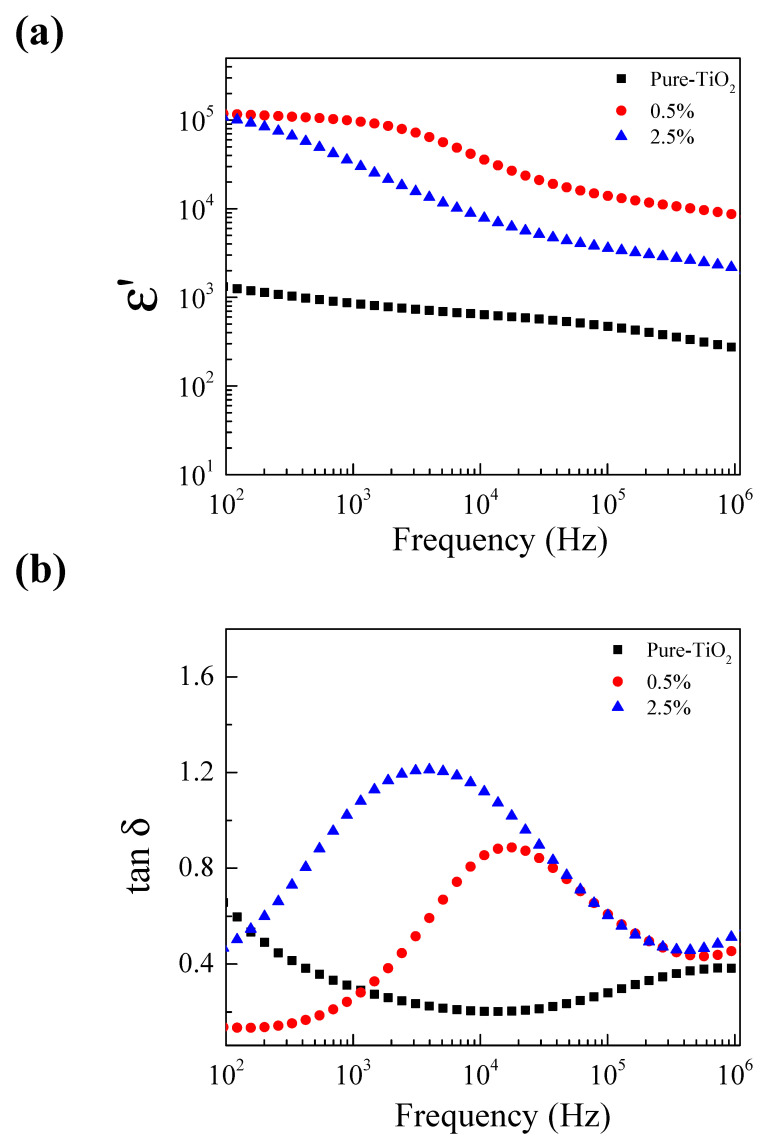
Frequency dependence of (**a**) ε′ and (**b**) tanδ at RT for ZNTO ceramics with various codopant concentrations (*x* = 0.5%, 0.25%).

**Figure 6 molecules-27-06121-f006:**
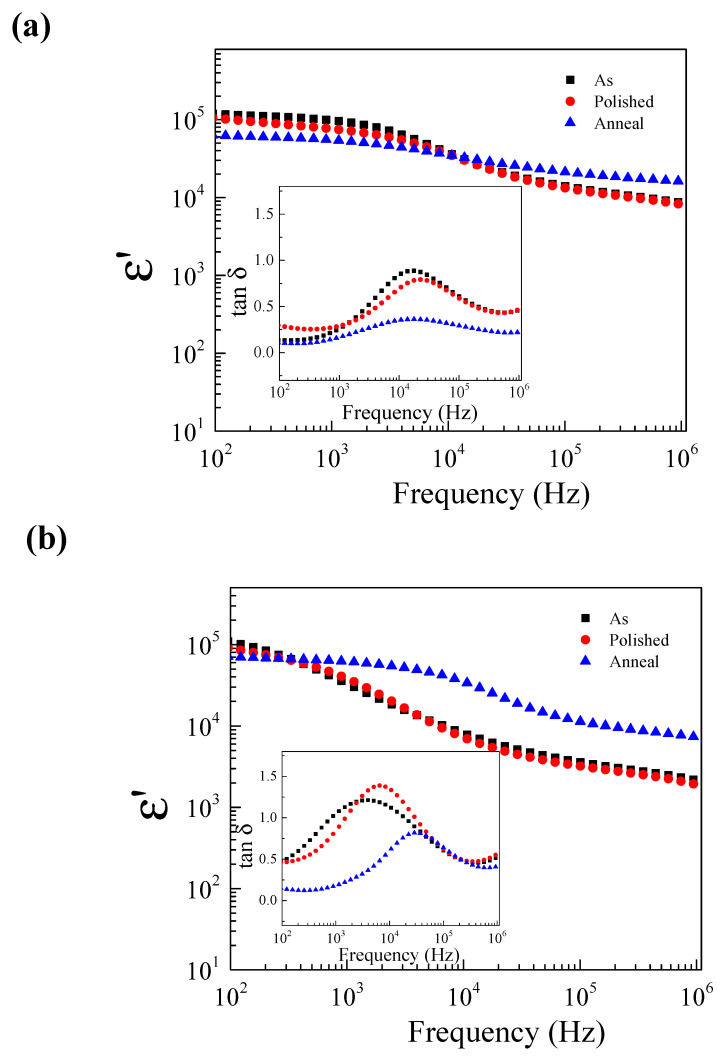
Frequency dependence of ε′ at RT of the as-fired sample, polished-sample and annealed-sample (in air) for (Zn_0.33_Nb_0.67_)*_x_*Ti_1−*x*_O_2_ ceramics with *x* = 0.5% (**a**), 2.5% (**b**); inset shows frequency dependence of tanδ at RT.

**Figure 7 molecules-27-06121-f007:**
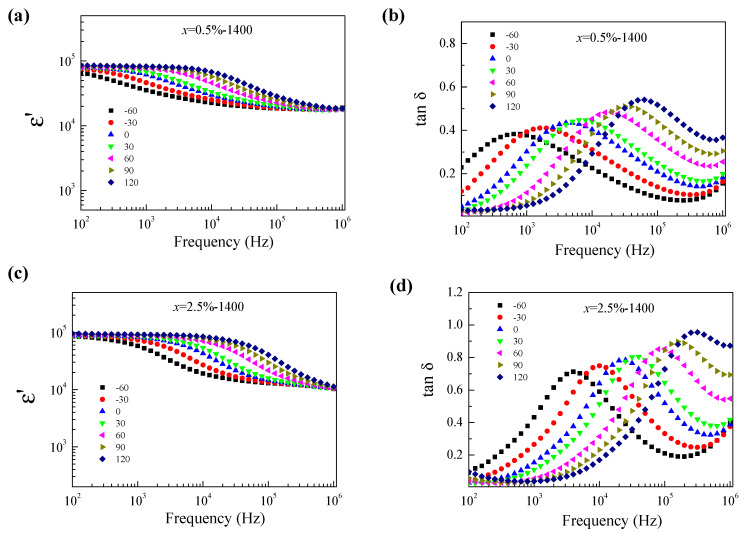
Temperature dependence of the (**a**,**c**) dielectric relaxation and (**b**,**d**) tanδ at different frequencies for ZNTO ceramics with *x* = 0.5 and 2.5% respectively.

**Figure 8 molecules-27-06121-f008:**
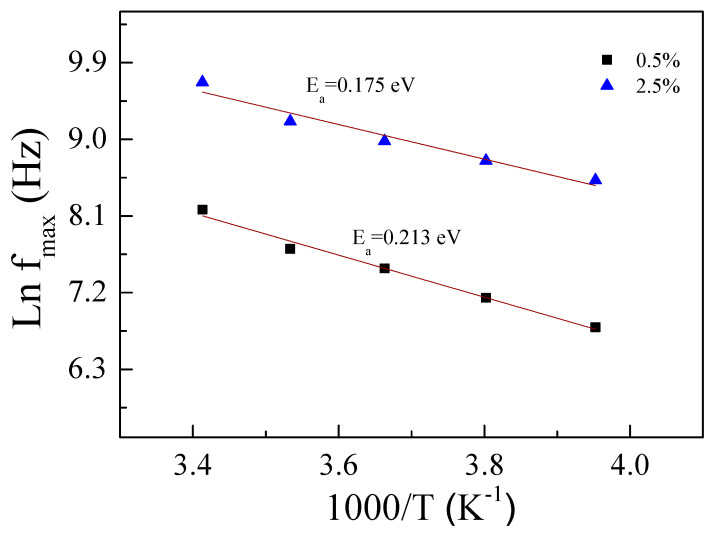
Arrhenius plots of the dielectric relaxation process for 0.5% and 2.5% ZNTO ceramics.

**Figure 9 molecules-27-06121-f009:**
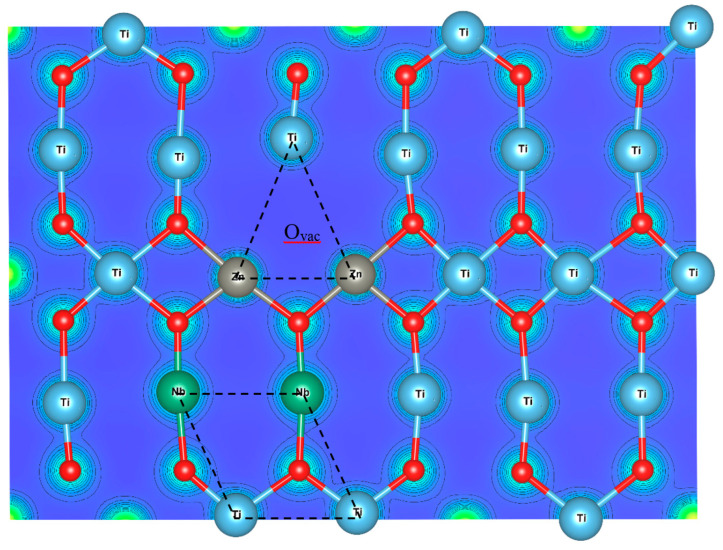
Energy-preferable structure of the 2ZnVo triangular defect and 2Nb diamond defect.

**Table 1 molecules-27-06121-t001:** Lattice parameter of ZNTO powders and ceramics with difference codoping levels.

Sample		Lattice Parameter (Å)	
		*a*	*c*
(a)	0.5% Zn-TiO_2_ powder	4.5961	2.9618
(b)	0.5% Nb-TiO_2_ powder	4.5964	2.9619
(c)	(Zn_0.33_Nb_0.67_)*_x_*Ti_1−*x*_O_2_ powder with *x* = 0.5%	4.5971	2.9621
(d)	(Zn_0.33_Nb_0.67_)*_x_*Ti_1−*x*_O_2_ powder with *x* = 2.5%	4.5994	2.9624
(e)	0.5% ZNTO	4.5967	2.9625
(f)	2.5% ZNTO	4.5997	2.9647

## Data Availability

The study did not report any data.

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
