# Peer review of "Dielectric Responses of (Zn0.33Nb0.67)xTi1−xO2 Ceramics Prepared by Chemical Combustion Process: DFT and Experimental Approaches"

_molecules, 2022, doi:10.3390/molecules27186121_

Round 1

Reviewer 1 Report

The paper autors  Theeranuch Nachaithong, Pairot Moontragoon and Prasit Thongbai "Dielectric Responses Of (Zn0.33Nb0.67)xTi1-xO2 Ceramics Prepared by Chemical Combustion Process: DFT and Experimental Ap-proaches " is devoted to the synthesis and study of the physical properties of TiO2 ceramics doped with a small amount of Zn and Nb. The work is relevant and interesting, and may be published in the journal "Molecules", after changes in accordance with the following remarks.

 - On page 2 in the second paragraph, the authors introduce the notation:

 (Zn0.33Nb0.67)xTi1-xO2 ceramics with x = 0.5% and 0.25% are referred to as the 0.5%ZNTO and 2.5%ZNTO ceramics, respectively.

Then in table 1:

 (e) (Zn0.33Nb0.67)xTi1-xO2 ceramic with x = 0.5%

(f) (Zn0.33Nb0.67)xTi1-xO2 ceramic with x = 2.5%

Then on page 4:

0.5%Nb+Zn codoped TiO2, and 2.5%Nb+Zn codoped TiO2 ceramic

The designations of the samples should be brought to the same form, understandable to readers.

 - The authors say that "... the Raman spectra of sintered ceramics were measured with a UV−vis Raman System." The System must be specified as UV-Vis and Raman are different measurement methods. The wavelength of the laser used to measure the Raman spectra should be given.

 - On page 4 it says “It was shown that oxygen vacancies and O–Ti–O bonds were associated with the Eg and A1g modes [16].” In [16], phonon modes were not studied.

 - The Arrhenius formula should be given in order for the reader to understand the notation (fmax, Ea).

 - DFT calculations are described very sparingly. It would be useful to inform readers how substitutional atoms and defects were taken into account in calculations in a periodic structure.

Reviewer 2 Report

1. The abstract needs a revision where the author was compulsory to include quantitative results.

2. The potential application needs to be added to the abstract.

3. The problem statement needs to be added in the introduction part.

4. Surface morphologies for all samples must be added and discussed in Fig. 2.

5. The elements mapping for all samples must be added and discussed in Fig. 4.

6. Frequency dependence for all samples must be added and discussed in Fig. 5.

Reviewer 3 Report

1. (Zn0.33Nb0.67)x Ti1-xO2 ceramics were prepared and investigated in this paper. On the top of page 2, the authors state that „Therefore, in this work the mechanic inside of the giant dielectric permittivity of TiO2-based ceramics will be reported and clarified.” However, the the mechanisms of giant permittivity for the investigated samples are not clearly explained.

2. The theoretical work is only shortly shown. Despite the claim in the Abstract that „.. it theoretically confirms that the electron-pinned defect-dipoles (EPDD) cannot be created in the ZNTO structure.” no argumentation is given in the manuscript.

3. In section 2 „Experimental details”, the authors specify that the dielectric measurements were carried out in the 102-106 Hz frequency range. However, in Figures 5 and 6 , data are presented between 50 Hz and 107 Hz. Please explain the discrepancy. In addition, please explain  whether or not the instrument KEYSIGHT E4990A  is of option 120 .

4. In order to provide the reader with a clear image on the experimental methods used, please5give some information on the instruments used for XRD, SEM, Raman.  

5. More attention should be payed to the editing of the manuscript. In the first line of the 1. Introduction instead „Recently, giant or colossal dielectric oxide materials…” should be „Recently, giant or colossal dielectric permittivity oxide materials…”

6. The section 4. „Conclusions” should be extended, by emphasizing on the deductions from the obtained results. For example, please discuss whether the annealing is useful or not. Again, the authors should pay attention to the English. For example instead „Moreover, their dielectric characteristics also are good frequency-stability…” should be „„Moreover, their dielectric characteristics show a good frequency-stability…”.

Reviewer 4 Report

In this work, the roles of (Zn, Nb) co-doping on dielectric properties of TiO2 ceramic are studied, and the authors provide a substantial experimental data and give a correct conclusion. However, my questions are as follows:

(1) An as-sintered pure TiO2 ceramic is suggested to present for comparison for the doped samples.

(2) Although the ZNTO ceramic shows a quite high permittivity that is necessary to storage devices, however, the dielectric loss is also very high which will limits its application. Some discussion should be given on the high dielectric loss of ZNTO.

(3) 2.5% is not a low doping concentration, bur no inpurity phases are checked in this work. The EDS results is not unpersuasive and spot scanning at grain boundary area or some impurity-like phases are suggested.

(4) What is the physical mechanism of the dielectric relaxation with activation energies of 0.213 and 0.175 eV? The value is variable with doping contents. Will it be related to the grain boundary imepedance?

Round 2

Reviewer 4 Report

The comments given in the last review have been addressed. I think it can be accepted after the Figure caption (in Figure 5) is revised. The pure TiO2 should be refered in the figure caption.